# “If You Do Not Take the Medicine and Complete the Dose…It Could Cause You More Trouble”: Bringing Awareness, Local Knowledge and Experience into Antimicrobial Stewardship in Tanzania

**DOI:** 10.3390/antibiotics12020243

**Published:** 2023-01-25

**Authors:** Jennika Virhia, Molly Gilmour, Cairistiona Russell, Edna Mutua, Fortunata Nasuwa, Blandina T. Mmbaga, Stephen E. Mshana, Torre Dunlea, Gabriel Shirima, Jeremiah Seni, Tiziana Lembo, Alicia Davis

**Affiliations:** 1School of Social & Political Sciences/School of Health and Wellbeing, University of Glasgow, Glasgow G12 8QQ, UK; 2KEMRI-Wellcome Trust Research Program, Kilifi P.O. Box 230, Kenya; 3Kilimanjaro Clinical Research Institute, Kilimanjaro Christian Medical Centre, Moshi P.O. Box 2236, Tanzania; 4Directorate of Research and Consultancies, Kilimanjaro Christian Medical University College, Moshi P.O. Box 2240, Tanzania; 5Department of Microbiology and Immunology, Weill-Bugando School of Medicine, Catholic University of Health and Allied Sciences, Bugando Medical Centre, Mwanza P.O. Box 1464, Tanzania; 6School of Biodiversity, One Health & Veterinary Medicine, University of Glasgow, Glasgow G12 8QQ, UK; 7Nelson Mandela African Institution of Science and Technology, Arusha P.O. Box 447, Tanzania

**Keywords:** antimicrobial stewardship (AMS), antimicrobial resistance (AMR), antimicrobial use (AMU), antimicrobials, antibiotics, healthcare providers, Tanzania, East Africa, sub-Saharan Africa

## Abstract

Antimicrobial resistance (AMR) is a global health issue disproportionately affecting low- and middle-income countries. In Tanzania, multi-drug-resistant bacteria (MDR) are highly prevalent in clinical and community settings, inhibiting effective treatment and recovery from infection. The burden of AMR can be alleviated if antimicrobial stewardship (AMS) programs are coordinated and incorporate local knowledge and systemic factors. AMS includes the education of health providers to optimise antimicrobial use to improve patient outcomes while minimising AMR risks. For programmes to succeed, it is essential to understand not just the awareness of and receptiveness to AMR education, but also the opportunities and challenges facing health professionals. We conducted in-depth interviews (*n* = 44) with animal and human health providers in rural northern Tanzania in order to understand their experiences around AMR. In doing so, we aimed to assess the contextual factors surrounding their practices that might enable or impede the translation of knowledge into action. Specifically, we explored their motivations, training, understanding of infections and AMR, and constraints in daily practice. While providers were motivated in supporting their communities, clear issues emerged regarding training and understanding of AMR. Community health workers and retail drug dispensers exhibited the most variation in training. Inconsistencies in understandings of AMR and its drivers were apparent. Providers cited the actions of patients and other providers as contributing to AMR, perpetuating narratives of blame. Challenges related to AMR included infrastructural constraints, such as a lack of diagnostic testing. While health and AMR-specific training would be beneficial to address awareness, equally important, if not more critical, is tackling the challenges providers face in turning knowledge into action.

## 1. Introduction

Antimicrobial resistance (AMR) affects people, livestock, and health systems biologically, economically, politically, and morally worldwide. While AMR is a global health challenge, it disproportionally affects low- and middle-income countries (LMICs) and particularly those in sub-Saharan Africa, who suffer the greatest impact of bacterial AMR [1]. Here, people experience high disease burdens while also facing inadequate sanitation and challenging health infrastructures, two critical issues that increase reliance on antimicrobial treatments [2,3]. Moreover, in these settings, people depend on livestock for their livelihoods and are therefore affected by AMR in multiple ways, both in terms of their own health and that of their livestock [4]. 

In order to address issues of AMR, formal training and guidance are essential, and these are critical factors in antimicrobial stewardship (AMS). AMS is widely advocated for in the global health community in order to optimise antimicrobial use to improve patient and animal outcomes while minimising AMR risks [5]. AMS therefore involves working with healthcare providers to ensure best selection and handling of antimicrobial treatment (e.g., dose, regimen, route and duration) and prevention of unnecessary use (e.g., prescription of antibiotics to treat viral or non-infectious conditions). Ideal AMS needs to be associated with access to affordable and quality-assured antimicrobials, infrastructure and interventions to prevent infection, hence antimicrobial use in, for example, water, sanitation and hygiene (WASH) in communities and infection prevention and control (IPC) measures in hospital environments. 

The World Health Organization (WHO) defines five pillars of AMS programming [5]. This paper focuses on pillar 3, which states the importance of “strengthening health worker capacity through the provision of tailored education and training packages according to health worker roles and functions.” Tanzania is addressing AMS through its fourth strategic objective in the National Action Plan (NAP) on AMR (2017–2022): “Optimize the Use of Antimicrobial Agents in Human, Animal and Plant Health” [6]. A key component of this objective is to “educate prescribers, pharmacists, nurses and lab personnel about good antimicrobial prescribing practices and antimicrobial resistance.” We also focus on strategic objective 1 of the NAP which is to, “Create Awareness and Understanding of Antimicrobial Resistance through Effective Information, Education and Communication.” We focus on this because an important component of AMS programmes is training on AMR, education, communication, and mitigation strategies in professional education. Small-scale AMR stewardship programmes have previously been conducted across Africa, with some showing positive improvements in prescribing habits. For example, in Kenya, the state created government-led guidelines tailored to their specific context that formed the basis of AMS programmes [7]. Similarly, in South Africa, a locally-relevant AMS programme was implemented in 55 hospitals that led to a 12.1% reduction in daily doses per 100 patient days within the first trimester of implementation [8]. In the veterinary and agricultural sectors, a number of systematic reviews have been conducted across varying contexts to highlight factors that enable effective AMS, including education, diagnostics, clinical guidelines, enhanced prescription checking, computer-based decision support, and strong leadership, accountability and responsibility [9,10]. Factors that were found to inhibit AMS include ineffective regulation (e.g., illicit drug store sales of antibiotics) and health system factors such as a lack of resources and clinical governance, as well as tensions between AMS and the need to maintain profitable business [11].

Evidently, while generic training may offer some beneficial outcomes, it is crucial that programmes also fit the needs of individual countries and health provider communities and cater to a variety of settings within a country, given complex health care circumstances. In addition, developing qualified human resources without sufficient consideration of opportunities and challenges of health systems and infrastructures that enable or prevent providers to incorporate knowledge into daily practices is unlikely to result in sustainable solutions. Finally, health provision in a sub-Saharan African context comprises a broad range of actors (see, for example, Table 1 and Table 2 which summarise health providers in our study context, Tanzania), and therefore AMS programmes need to be tailored accordingly. Veterinary health provision is especially under-represented in the overall AMS discourse, and particularly in sub-Saharan Africa and LMICs more generally [12].

In this paper, we examine these issues in a northern Tanzanian context. Here, antibiotic use is prevalent in both community and clinical settings [13,14,15,16]. In clinical settings, antibiotics are often used empirically to treat a likely bacterial infection in children under two years of age and in patients admitted to surgical and paediatric wards [14]. The prevalence of multidrug-resistant (MDR) bacteria is high [17,18,19]. For example, about two-thirds of isolates from wound infections at Muhimbili National Referral Hospital were found to be resistant to at least three classes of antimicrobials [20]. In another study at Bugando Medical Centre (BMC), the prevalence of MDR-Gram-Negative-Bacteria (GNB) isolated from blood culture and GNB third-generation cephalosporins isolated from neonatal colonization, cots and mothers’ hands were 96.6, 100, 100 and 94.6%, respectively [21]. Additionally, a study found that MDR uropathogens caused community-acquired urinary tract infections in patients attending health facilities in Mwanza and Dar es Salaam [22]. In animals, *E. coli* isolates from poultry and pigs were found to be highly resistant to tetracycline (63.5%), nalidixic acid (53.7%), ampicillin (52.3%), and trimethoprim/sulfamethoxazole (50.9%) [23]. Antibiotic use is also high in community settings, both in livestock and in people [12,24,25,26]. Patients and livestock keepers obtain antimicrobial treatment for themselves and their animals either through health providers in the human and veterinary health systems or through self-access, e.g., via accredited drug dispensing outlets (ADDO) and market vendors [27,28]. Self-access is popular in part because systems are strained, underfunded, or rely on private-public formats that are absent or expensive [29]. It is not only patients and livestock owners who face challenges around AMR – health providers themselves (human and veterinary) are often implicated as key drivers of AMR through their prescribing practices, which are a product of constrained systems of care [30].

In Tanzania, access to professional education and specialised training for health providers is challenging due to a limited number of higher education institutions that are able to provide this. Access to healthcare and other infrastructure for patients and livestock owners is also severely constrained. In this paper, our definition of “healthcare providers” includes both biomedical and traditional providers, for example, doctors, medical officers, veterinarians, livestock field officers, community (animal) health workers, medicine vendors, and traditional healers (see Table 1 and Table 2). The role of health providers and broader medical care in Tanzania spans a wide range of health facilities, structures, and institutions, all of which employ health practitioners with varying degrees of education, qualifications and experience [31,32,33]. These sources of medical care can be grouped into the following categories, spanning from local to regional and zonal scales: those who work within established heath institutions such as qualified medical doctors, clinical officers, nurses, veterinarians and livestock extension agents; those who work within the market-based economic sector, such as farmer market’s sellers, pharmacies and agrovets; and those who are primarily community-based, e.g., community animal health workers (CAHWs), community health workers (CHWs), and local and traditional healers alongside farmers. Health practitioners within these institutions have a range of training and expertise (see Table 1 for human health providers and Table 2 for veterinary health providers). At the local level, health infrastructures can include rural health clinics, traditional health provision, dispensaries, ADDOs and “informal” medicine shops. District, regional and zonal health facilities, namely hospitals, serve broader catchment populations and have access to better infrastructure, although constraints exist also at this level [30].

**Table 1 antibiotics-12-00243-t001:** Human health providers in the Tanzanian health system.

Level	Human Health Role	Description	Training	Recognition	Ability to Dispense/Sell Antibiotics
Health institutions: Degree holders and mixed post-secondary qualifications/training/knowledge	Medical Doctor (MD)/Officer	Most MDs are in the district hospitals, with a few in Health Centres. In district and remote rural communities, they treat patients with mild or chronic illnesses and refer those with serious conditions to higher-tier hospitals [34].	Degree in medicine	Medical Council of Tanganyika	Yes
	Clinical staff (Assistant Medical Officers, Clinical Officers, Assistant Clinical Officers)	Deliver the majority of care at district level and in rural primary health facilities (dispensaries and health centers [34]).	Four to six years of secondary education followed by three to four years of professional training at certificate and diploma levels.	Medical Council of Tanganyika	Yes
	Nurse (Nursing Officers, Assistant Nursing Officers, Nurse Mid-wives, Public Health Nurse)			Tanzania Nursing and Midwifery Council	Yes
Retail/Market-based providers	Type I drug providers	Sell prescription-only medications (POM), including antibiotics, pharmacy only (PO), and general sale list (GSL), medicines [16]. Must be run by registered pharmacist.	Degree in pharmacy	Pharmacy Council Tanzania, Tanzania Medicines and Medical Devices Authority (TMDA)	Can sell antibiotics with prescription
	Type II drug providers e.g., Accredited Drug Dispensing Outlet (ADDO*)	Can only sell medicines on the GSL (e.g., common painkillers, cold and flu remedies), and some antibiotics with prescription, e.g., amoxicillin capsule/suspension, benzyl-penicillin powder for injections, chloramphenicol eye drops, trimethoprim/sulfamethoxazole suspension, doxycycline capsules/tablets.	Staff required to have 4 years minimum training e.g., pharmaceutical technician or nurse at diploma or certificate level.	Pharmacy Council Tanzania, Tanzania Medicines and Medical Devices Authority (TMDA)	Can sell some antibiotics with prescription
	General stores/community shop	Sell general supplies plus some over the counter drugs. Do not require registered pharmacist.	No professional certified training	Community-based and government	Cannot legally sell antibiotics
Community health providers	Community Health Workers	Health promotion and basic curative and preventative services. Can advise sick people to go to a healthcare facility [35].	Secondary education and short health training course	Community-based and government	Cannot legally sell antibiotics
	Traditional Healers	Treatment often consists of a combination of ritual and herbal medication, offering extra, specific treatment for protection of body and mind that cannot be treated in hospital.	Through family and peers	Community-based and some are registered with the Traditional and Alternative, Medicine Council.	Cannot legally sell antibiotics

* The Tanzanian Accredited Drug Dispensing Outlet, or ADDO, was established by the Pharmacy Council in 2009 to improve access to essential medicines, including antibiotics, by implementing trainings and inspections. According to Tanzanian regulations, antibiotics can only be dispensed with a prescription from a doctor at an ADDO dispensary. However, studies have shown that unregistered ADDO pharmacies and other shops continue to sell antibiotics without a prescription, with some research concluding that this dispenser behaviour is motivated by the need to make a profit and meet customer demand [35,36].

**Table 2 antibiotics-12-00243-t002:** Veterinary health providers in the Tanzanian health system.

Level	Animal Health Role	Description	Training	Recognition	Ability to Sell/Dispense Antibiotics
Health institutions: Degree holders and mixed post-secondary qualifications/training/knowledge	Veterinarian	A person qualified to treat diseased or injured animals.	Degree in veterinary medicine	Veterinary Council of Tanzania	Yes
	Paraveterinarians (including Livestock Field Officers, LFOs)	Work under the supervision of a veterinarian. Provide services ranging from disease surveillance to artificial insemination, animal treatment, vaccinations, and extension services, among others [37].	Certificate or diploma holders in animal health	Veterinary Council of Tanzania (if registered)	Yes under veterinary supervision
Retail/Market-based vendors	Agrovets/livestock medical vendors/veterinary centers	Retail livestock stores supplying basic drugs, animal feed, seed, and fertiliser to farmers. Key source of animal health advice.	Short training course	Veterinary Council of Tanzania (if registered) TMDA & Tanzania Plant Health and Pesticides Authority (TPHPA)	Yes
	Open-market livestock drug vendors (livestock auction)	Street vendors working in informal economy, selling from open air courtyards and streets.	No professional certified training	Community-based	Cannot legally sell antibiotics
Community health providers	Community Animal Health Workers	Selected by their communities and trained in the prevention or treatment of a limited range of animal health problems [38].	Secondary education and short training course	Community-based	Cannot legally sell antibiotics
	Traditional Healers	Treatment often consists of a combination of ritual and herbal medication, offering extra, specific treatment for protection of body and mind that cannot be treated solely through biomedical approaches.	Through family and peers	Community-based	Cannot legally sell antibiotics
	Expert Farmer	Practices agriculture, often keeping and caring for cattle and poultry for personal use rather than industrial commercial sale.	Through family, peers and LFO	Community-based	Cannot legally sell antibiotics

In this study, we explore health providers’: (1) motivation to engage in health matters and broader roles in community health; (2) technical training; (3) awareness, knowledge and perceptions of infectious diseases and AMR, and practices contributing to AMR; and (4) constraints in these daily practices, including those within and beyond the health system. By examining these four themes, we provide a holistic overview of the experiences of health professionals in northern Tanzania in relation to AMR, and of the contextual factors surrounding their practices that might act as enablers or barriers to the translation of knowledge into action. Given the focus of education, communication, and awareness in both the WHO and NAP strategic priorities for AMR and AMS, we highlight key opportunities and challenges, alongside identifying training needs that would enable or compromise the design and delivery of antimicrobial stewardship programmes in the region. While this paper brings attention to education-based issues (such as knowledge and awareness), we do so while recognising how these critical components of AMS are situated in broader health landscapes, contexts, constraints and capabilities of individuals, and the health system more broadly. In doing so, we move beyond typical knowledge attitudes and practice (KAP)-based studies to explore broader experiences of health and health practice [39].

## 2. Methods

### 2.1. Data Collection—Sites and Participants

Data were collected during April–May 2019 and September 2020–February 2021 through 44 in-depth interviews (IDIs) and 18 focus group discussions (FGDs) in six villages in three regions (Kilimanjaro, Arusha and Mwanza) of northern Tanzania. The villages are representative of key livelihood strategies predominant in rural East Africa (agro-pastoral, pastoral and rural smallholder). Villages were selected based on human and livestock population sizes, number of sub-villages and access to healthcare in the form of hospitals and/or dispensaries, veterinary offices and drug shops. Participants were purposefully selected in order to provide representation of each type of health provider in each study location. The sample size was capped once no new themes emerged from the data. The results presented in this paper draw from analysis of IDIs only (see Appendix A for an extract of interview questions). However, FGD data were used to inform the inductive coding framework and provide general contextual understanding. Table 3 outlines the number, location, and health provider type of participants.

### 2.2. Interview Protocols

Interview schedules covered a number of themes, including personal motivations and experiences working within the health system; training in health-related matters, including AMR; questions about AMR knowledge and attitudes; diagnostic processes (e.g., whether diagnostic tests are available and waiting times to receive test results); treatment processes, including practices of self-treatment in communities; biosecurity and infection, prevention and control practices; and challenges faced both within and outside the health system.

All interviews were conducted or moderated by Tanzanian research assistants fluent in the main languages spoken in the study locations, Swahili or Maa. Interview protocols were developed to steer the discussion but retained a degree of flexibility in order to capture the natural flow of the conversation. Notes were taken throughout the IDIs by dedicated note-takers and the conversations were also audio-recorded. All data were collected following strict ethical protocols and approvals via the University of Glasgow, the National Institute for Medical Research in Tanzania (NIMR) and with verbal and/or written consent from all participants. Details of our ethical protocols can be found in an ethics statement in the acknowledgements.

### 2.3. Data Analysis

Interviews were recorded (when consented to), transcribed and translated to the English language by Swahili and Maa speakers fluent in English. Translation occurred in order to render meaning from Swahili/Maa into English. Interviews were de-identified, and anonymised with identifying information removed before translation. Translations were initially word-for-word, with a follow-up interpretive approach to make them readable and understandable in standard British English. This was done as in some cases the Swahili wording was translated literally which elicited little meaning in English. The follow-up interpretive approach took into consideration the broader context of the translation, thus allowing the analysts to interpret the intended meaning of the translation. Once interviews were translated, they were reviewed by the field team lead, field coordinator, and project Co-Is. They were checked for errors and segments that were not well translated were re-retranslated and/or clarified. Quality assurance was done through multiple reviews of the transcriptions and translations by multiple parties. Any final clarification or grammatical ‘cleaning’ has been done for quotes used in the text to ensure they are comprehensible in standard British English.

The interviews were analysed using NVivo 12 (QSR). A coding matrix was built from a mix of inductive codes (deriving from the data themselves) and deductive codes (informed by research questions and literature) and applied across the data sets. An association matrix between individual codes was used to group them into broader themes. Further review of the literature contextualised these emergent themes. Using a mixed inductive and deductive coding approach is common in qualitative data analysis, as it incorporates ideas from pre-existing literature to ‘ground’ the research while also representing the realities of participants through themes that emerge through the analytical process [40].

## 3. Results

We present data below as they relate to the four key areas affecting AMS listed above, namely: (1) motivation to engage in health matters and broader roles in community health; (2) technical training; (3) awareness, knowledge and perceptions of infectious diseases and AMR, and practices contributing to AMR; and (4) constraints in these daily practices, including those within and beyond the health system.

### 3.1. Motivation and Role in Community Health

Interviewees reported two primary motivations for working in healthcare: “supporting the community” as well as supporting their own and community “livelihoods” (see Table 4 and Table 5). “Supporting the community” was the most reported motivation, primarily expressed by interviewees working in human healthcare in their communities. In particular, Community Health Workers (CHWs) stated that their most important duties relate to educating and advising community members on health-related matters, including infection prevention protocols, for example, boiling water and milk, using nets for protection against mosquitoes, and basic personal hygiene practices such as hand washing or toilet use.

*“[First], I am happy because we are helping to conserve the environment, keeping [the community] neat and clean because sometimes you might go to some places and find out someone has no toilet, or his toilet is full and overflowing. So, what we do is to educate them on the importance of toilets and how to maintain them and the environment, as well as the importance of washing hands.”*—Community Health Worker 4, Mwanza

They also encourage patients to seek medical attention rather than administering treatment. They explained they were not being paid in their roles and that they secured these posts in community elections. A CHW in Kilimanjaro explained: 

*“…frankly the only thing that makes me happy is because I am volunteering, and people are getting their rights and they go to the hospital without much trouble, but the problem in the work [lack of pay] itself makes me not enjoy working.”*—Community Health Worker 5, Kilimanjaro

Helping their community was also expressed as a motivation by paid clinical practitioners including nurses, medical officers and livestock officers. A nurse practising in Mwanza described their motivation for working in healthcare as: 

*“What makes me most happy is [when] a person comes in sick […] you ask him how he is doing […] then you see him doing completely well […] I feel completely at peace that what you are doing brings good results even for patients.”*—Nurse 1, Mwanza

Some healthcare workers also described their role in educating and communicating to patients the increased risks of AMR and drug resistance, and the need to complete a prescribed dose: 

*“I usually tell you for example, if you do not take the medicine and complete the dose… that really causes drug resistance, so you tell them it could cause you more trouble.”*—Nurse 2, Mwanza

However, there was great inconsistency in reported answers when people were asked who is ultimately responsible for preventing antimicrobial resistance.

A second commonly-reported motivation for working in healthcare was supporting the livelihoods of communities as well as their own. Farmers and drug dispensers expressed the importance of the income that these roles generate. A Livestock Field Officer in Mwanza outlined the liquidity that keeping cattle brings to cattle owners which cannot be sourced through other measures: 

*“…pastoralism is a very valuable thing because now … I have sold one bull… one million and a half. [TSH/$650 USD] I have been able to… buy blocks and now the house is at [has been built to] its lintel. […] Even at the bank you cannot be given [money quickly], but if I sell the cow, which is [to be] consumed by people, maybe five or six hundred [thousand TSH/$250 USD], you are given money the same day. Therefore, you get rid of your problems, the cow is a thing which removes your problems at once.”*—Livestock Field Officer, Mwanza

Farmers explained that they generated income by selling offspring, using manure from livestock to fertilize crops to improve yield, and one interviewee described that he kept livestock “for having milk and food in case of harder times” [Expert Farmer 2, Kilimanjaro]. There were cases of retail drug vendors having formerly worked as nurses or animal health specialists who began selling medicines to support their livelihoods in response to the high demand for medications in their community. In summary, the two primary motivations for healthcare workers were found to be their passion, described as a calling, to help and teach their local communities in healthcare, alongside the livelihood stability that some of these roles can offer to individuals and communities. 

### 3.2. General Education and Technical Training 

There was a diverse range of training reported by the healthcare professionals interviewed in this study, from university degrees to technical training certificates, short courses and receiving guidance from family and friends alongside self-study using textbooks and pamphlets. 

Expert farmers and traditional healers often reported having received informal training from peers and older family members. A farmer in Kilimanjaro described their training as follows: 

*“I was trained a bit by a Livestock Officer who gave me a book which explained the way cows should be kept, when to bring in a male for breeding, and what types of grass we should feed them at what time. I once lent [the book] to one of my friends but unfortunately […] my book got lost. There was a diagram with a scale which showed if you want to sell the cows, how much they should weigh for a certain amount of meat. That is what I have forgotten but the rest of the book’s content I can remember.”*—Expert Farmer 1, Kilimanjaro

Community health workers repeatedly reported receiving on-the-job training ranging from two days to three weeks. A community health worker in Arusha describes a typical on-the-job training: 

*“I: What training do you have as a community-based health worker? R: I don’t have much training. I: Is that just a seminar? R: It’s just a seminar, and there are times when I’m given a seminar, two days we got there at [name of location], but we don’t have much training. I: What did you learn in that two-day seminar? R: We were taught how to visit the public to get their information so that we can bring information to the health facility.”*—Community Health Worker 2, Arusha

Nurses, medical officers and animal health specialists often reported having two to three years of college and professional training. Many interviewees explained that they turn to literature like leaflets and textbooks alongside colleagues for further guidance. Community health workers and human drug vendors exhibited the broadest range of training, from informal on-the-job training to diploma or certificate level. The duration also varied from days to weeks. Livestock medical vendors had primarily received informal guidance from peers and family. Regarding specific education and training surrounding AMR, most of the respondents had not received specialised training. Participants on the whole reported having some form of training, whether formal or informal, for their role, but the majority of participants agreed that they would be interested in receiving more formal training in healthcare relating to antibiotic use and resistance. 

*“I: Have you got any training or report concerning antibiotic drugs becoming resistant? R: Not yet. I: Would you want to get training? R: I would want to get, if it is, or becomes available, I would like to get that training, I would even use it to sensitise the community.”*—Community Health Worker 1, Mwanza

### 3.3. Knowledge and Awareness of Infectious Conditions and AMR

#### 3.3.1. Practical Knowledge of Common Infections 

When asked questions surrounding bacteria, viruses, and the causes of common infections found within the providers’ area of practice, several misperceptions were apparent, particularly among the providers who had not received specific training or had no formal education. For example, a CAHW identified the causal agent of anthrax as a virus, while a CHW suggested that malaria could be transmitted sexually. Another CHW, who received a one-year training course on women’s health and pregnancy, said it was common for women to acquire urinary tract infections from stepping in unclean water or on soil that had been contaminated with urine.

However, other providers were able to clearly articulate their understanding of common bacterial and viral infections. For example, three CHWs, and a livestock medical vendor, identified major bacterial and viral infections affecting their communities. These included HIV, malaria, tuberculosis, urinary tract infections and respiratory infections. Two CHWs also identified the source of some of these infections, for example contaminated milk and water, as demonstrated by these quotes:

*“Now something that causes this stomach disease most of the time [is not] fetching water in the well! Now other time[s] we [ingest] those bacteria in [that] water because we do not boil.”*—Community Health Worker 2, Mwanza

*“That is why when you get fresh milk, you have to boil it. [A] long time ago people didn’t know that, so they got sick.”*—Community Health Worker 1, Mwanza

#### 3.3.2. Knowledge of AMR and Its Drivers

Levels of knowledge and training on issues relating to AMR specifically also varied among health providers. Despite the limited specialised training highlighted above, most medical officers, nurses and livestock officers could articulate their understanding of antibiotics and AMR. In particular, healthcare providers demonstrated their awareness of issues relating to drug resistance and diseases that become difficult to treat. 

*“[Resistance] has been the case with anti-malaria drugs, we started with chloroquine, then came amodiaquine, then... sulfadoxine, now we have ALU [artemether-lumefantrine]. We also had quinine, which is also now only rarely used, so it’s like that, the same thing that happened with malaria can also happen.”*—Assistant Medical Officer, Mwanza

However, AMR was not always understood in relation to disease, particularly among community-based health providers. For example, in the case below, a participant alluded to drug resistance by expressing concern that medicines nowadays do not seem to work as well as they did in the past:

*“I: Ask him when he mentions current and past medicines, has performance been different? R: Of course, it’s different because the old medicine used to treat it well and it wasn’t a lot of medicine. And… when we are told that this medicine is good for cattle you just take it, so right now we see these drugs are a challenge again because they are so many drugs and we see it even destroying cattle, sheep, goats. The old ones are good because they were few.”*—Expert Farmer, Arusha

Regarding contributors of AMR, health providers often pointed to the actions of others, either patients or other providers, as key drivers. We discuss these in more detail below.

#### 3.3.3. Actions of Patients

Providers stated that incomplete dosing is an important driver of AMR and that end users are often responsible for this. Specifically, interviewees were generally able to demonstrate their awareness of the importance of completing prescribed doses, although some explained the difficulties of ensuring patients/customers followed these instructions. In particular, interviewees reported that patients often do not accurately take their dose as prescribed and often stop when they begin to feel better, which contributes to the development of AMR. 

*“R: Once they feel unwell they do run to the pharmacies to purchase all sorts of drugs and start using them but once they get better, they will stop or not finish the dose and this contributes to resistance.”*—Assistant Clinical Officer 3, Kilimanjaro

However, some try to articulate their concerns about incomplete doses to their patients:

*“R: I usually tell you [the patient], for example, if you do not take the medicine and complete the dose, that really causes drug resistance. So you tell them [the patient], ‘It could cause you more trouble [later]’.”*—Nurse, Mwanza

Health providers also cited patients treating themselves with medicines that they buy from pharmacies or markets as a key contributor to AMR. Providers believe that patients have become accustomed to treating themselves and therefore see no need to seek advice from healthcare staff regarding which medicines are appropriate for their condition. 

*“R: This is why I said earlier that people are so much used to these medicines. They simply buy and they know how to use them as they normally do. If you tell them: “Use it this way or that way”, they’ll tell you: “Don’t teach me. I will do it myself”. There are those who buy medicine and would like to be told how much cc to administer and some don’t want to know from you, saying that they normally do it by themselves.”*—Animal Medical Vendor 2, Arusha

Biomedical health care providers, particularly nurses and medical officers, often allocate blame towards patients for using traditional healers for issues relating to health as a potential contributor of AMR. Healthcare providers argued that traditional healers were trusted and favoured, with biomedical treatment being a last resort. One medical officer explained the preference for traditional medicine in their community: 

*“Education should be provided by traditional healers because they are highly trusted and that is why their rooms are filled with patients as more prioritise going there than coming to the hospital.”*—Medical Officer, Mwanza

The medical officer goes on to state that the preference for traditional healers could be capitalised upon and incorporated into the referral system: 

*“So I think education should be provided on the referral system because many referrals to us could be made by the traditional healers, to come to the hospital at early stages of their conditions.”*—Medical Officer, Mwanza

Finally, livestock field officers described preventative measures that can be taken, including vaccination campaigns, but argued that some community members would not implement all of them as some animals are favoured over others.

#### 3.3.4. Actions of Providers

Higher-tier health professionals often attributed the spread of AMR to the actions of other health providers. Medical officers and nurses were likely to blame retail vendors in particular, such as pharmacists and animal medical vendors, for high use of antibiotics in their communities, describing their pursuit of profit. Market vendors justified these actions by arguing that antibiotics are their best-selling products and they would lose customers and therefore income if they failed to meet customer demand: 

*“R: Antibiotic are the ones that sell the most and are very well known by my customers who are mostly livestock keepers… So, without antibiotics, I will miss a lot of customers.”*—Animal Medical Vendor 1, Arusha

In addition, they described them as capitalising on economic constraints by substituting costly medications with those that were more affordable for clients: 

*“So they switch the prescription, giving the patient a less costly medication, justifying that this is what your money can buy.”*—Medical Officer, Mwanza

Medical officers, nurses and animal health specialists were largely in agreement that antibiotics are widely over-prescribed, even if most did not identify their own prescribing practices to be implicated in this:

*“Antibiotics which are in use in veterinary medicine are broad spectrum antibiotics, therefore it is easy to find them prescribed for almost every complaint. That is why they are in regular use. I: Are they being overused? R: Yes.”*—Animal Health Specialist, Kilimanjaro

### 3.4. Constraints in Daily Practices

Independent of the training received or perceived levels of knowledge, there were a broad range of constraints expressed by interviewees that often made it challenging for them to perform their duties. Such constraints included poor physical infrastructure and other types of infrastructure, including diagnostic capacity, remoteness, poverty, staffing, and poor access to treatment. The providers linked these obstacles and the resulting practices to the development of AMR. We explore more specific links between these constraints and AMR below (summarised in Figure 1).

#### 3.4.1. Infrastructure

Poor physical infrastructure affected community health workers in particular, as they are often expected to provide health services for large areas and for a large number of clients. This is especially pronounced in pastoral rural areas where services are scarce, yet lack of roads and infrastructure often renders more remote areas inaccessible to providers. In many cases, providers do not have access to a vehicle and are required to visit clients on foot. A further challenge is that farmers often work away from homesteads and might not be on farms at the time the provider visits. Furthermore, community-based health providers often have to pay out of pocket for transport costs in order to reach their clients, thus affecting their ability to provide for themselves. 

Other infrastructural constraints include low staffing levels, insufficient remunerations and lack of training. A lack of time caused by staffing levels and a lack of resources can lead to poor record-keeping which can also result in over-prescribing in individual cases: 

*“I: Okay, how does time limits affect how you interact with patients? R: Time limit in taking patient history? I: Yes. Has it ever happened that you lacked sufficient time to interact effectively? R: Yes, it can happen, especially when we have staff shortage.”*—Assistant Clinical Officer 3, Kilimanjaro

Interviewees described the absence of resources needed for the role as a demotivating factor, restricting their ability to provide the care they aspire to give. In one case, a Community Health Worker explained that they “*gave up months ago*” [Community Health Worker 1, Kilimanjaro], as they had not received any funding for their role.

#### 3.4.2. Diagnostic Capacity

Another issue identified by healthcare providers was the lack of access to testing resources. As a result, they often solely rely on patient reporting and history to diagnose an illness and decide on a treatment plan. For example, one doctor described the overlap in symptoms between malaria and kidney infections, and how the lack of testing facilities can cause misdiagnosis and incorrect treatment. Health providers explained how their inability to correctly diagnose specific conditions and therefore inform their and the patient’s treatment choices could lead to AMR. For example, a health worker from Mwanza stated: 

*“R: “First, we don’t have many testing equipment like for urine and stool for UTI and worms, respectively. At the moment we rely on questioning to know what’s wrong with a patient because you can’t tell him/her to go and get the test done, then come and get the medicines. So, our challenges are the testing equipment. There is a scarcity of the most important equipment.”*—Assistant Clinical Officer 3, Kilimanjaro

This lack of diagnostic capacity may cause them to contribute to AMR by: 

*“R: …prescribing and dispensing a wrong drug, simply because you have asked a patient a question and they have answered you in a certain way. It could be a normal cough which could have been settled with a cough syrup.”*—Assistant Clinical Officer 3, Kilimanjaro

*“I: As a healthcare worker, do you think your personal behaviours can contribute towards the microbial resistance? R: Yes. I: How? R: I need to check and do tests on her in order to know what medicine to prescribe. I: Mmh, okay. R: I tell them, when you are ill, don’t just go to the pharmacy and purchase drugs, come and get checked what is wrong with you and once we prescribe something, make sure you finish the dose.”*—Assistant Clinical Officer 3, Kilimanjaro

With limited testing facilities for many healthcare providers, record-taking is particularly important and recognised by both animal and human health providers, as an animal health specialist states:

*“If it wasn’t me who was providing treatment previously in that area, I would ask a lot of questions regarding previous medical records. I would like to see them if they are available before I decide on the course of action if the disease is repetitive. If so, we can change the medication.”*—Animal Health Specialist, Kilimanjaro

#### 3.4.3. Drug Availability

A further constraint expressed among providers of all levels related to challenges in accessing and procuring medications. In the case below, an animal medical vendor describes the problem with purchasing drugs, particularly when retailers themselves rely on sourcing medications from third-party suppliers elsewhere in Tanzania or in Kenya. When met with the unavailability of medications, the vendor describes how they were required to abandon their planned course of treatment: 

*“I: Do you face any challenges when you go buy the medicines, if so, which ones?... R: The challenge that I normally face is that… there can be a certain medicine that I need depending on a disease and the needs of my customers. I: Yes. R: But when you get to town you might find that the medicine is not available and you may be told to wait until it’s ordered from Nairobi. I: Aha. R: So [I] would just have to come back and abandon the idea, because you cannot wait until it is ordered. I: Okay. R: Yes.”*—Animal Medical Vendor 1, Arusha

The vendor further explains that even when medications are available, they often arrive in poor condition, due to the way in which they are packaged and stored during transit: 

*“I: Okay. Another challenge? I: Yes… the other challenge… it’s transport. I: Yes? R: …sometimes they do not seal the boxes well… so they may arrive with a few broken bottles… That is when you send them using the bus… you must be present so that you seal them… They sometimes arrive with a few broken bottles, for instance, the dip medicine. I: Yes. R: Yes, the dip medicine container breaks a lot. I: I see. R: Yes. Sometimes you’ll see that. I: Yes. R: Once it’s broken, it ends up messing other things that are in the box. So sometimes the buses refuse to take them. I: To take them? R: It’s because once they spill, it’s poisonous. I: I see. R: They’ll tell you to hire a lorry for such medicine.”*—Animal Medical Vendor 1, Ngorongoro

Drug procurement was also an issue in the human health sector, where drugs are supplied by an autonomous branch of the Ministry of Health responsible for procuring and supplying drugs to health centres and hospitals: 

*R: “Another thing is the lack of drugs… The government says it will bring drugs since the year before last but they don’t bring any, only the verticals which are free like the contraceptives and syringes, but the actual drugs needed are not available.”*—Assistant Clinical Officer 3, Kilimanjaro

## 4. Discussion

The ultimate purpose of this research was to examine the experiences of health professionals in northern Tanzania in relation to AMR, and to assess the contextual factors surrounding their practices that might create opportunities or challenges in supporting the translation of knowledge into action. We targeted a community of human and animal health providers in northern Tanzania, representative of a range of rural agricultural settings in sub-Saharan Africa. We explored their motivations and technical training, understanding of infectious diseases and AMR, and practices contributing to AMR, as well as the constraints (individual and systemic) they face in their daily practices. We found that health providers were motivated and dedicated to supporting their communities of patients and farmers. There were many cases of people having experience in multiple roles, which suggests a broad understanding of different aspects of healthcare, ranging from community education and treatment to drug dispensing. However, levels of training and understanding of AMR varied, particularly among community health workers and retail drug dispensers. Factors perceived as contributing to AMR included the actions of “others”, e.g., patients or other health providers, as well as broader infrastructural inadequacies, such as limitations in drug supply chains and diagnostic capacity to inform treatment decisions. We conclude that, while critical, tailored training in health-related subjects, and AMR and AMS more specifically, would be insufficient on its own in addressing AMR in sub-Saharan Africa. Equally essential is understanding and tackling the practical challenges faced by healthcare providers that hinder their ability to utilise this knowledge, such as communicating AMR and building trust between providers and community members.

Our analysis reveals several gaps in the training health providers should receive and what their self-reported training was. This is consistent with other studies (see: [41] and [42]), which revealed high variation in levels of professional education among the health workforce across both low, middle and high-income country contexts. Further exploration of the difference in the state’s stipulated education requirements and actual training received across the healthcare sector is needed. Formal training in AMR and AMS more specifically was especially lacking. Overall, there was an interest in receiving further training across all professions in line with the great motivation expressed by interviewees to support community health on multiple grounds. These included their moral obligation, a genuine interest in medicine, and the desire to have the skills needed to help people in their community. 

Especially limited were formal training opportunities available to the community healthcare providers (veterinary and human). These providers are often deployed to meet the need created by an overstretched health workforce and to support the availability of, and access to, basic health services, especially in rural areas [43]. They are important contributors to overall healthcare because they have a close understanding of their communities, including local languages, cultural norms, and neighbours’ and families’ life and health experiences. They can reach households in very remote areas, can advise on personal hygiene and disease control measures, and have overall effective communication (often on a personal and knowing level, as trust and demonstrations of care are essential). Despite playing a critical role, however, they are often unpaid paraprofessionals or lay individuals, many of whom receive informal, short-duration, job-related training compared to other health professionals [43]. As such, they are often not held in the same regard as licensed medical professionals [44] and are not permitted to administer treatment. Studies are warranted to understand how these issues might impact their sense of agency around health and AMR matters affecting communities. Specific training tailored to this particular group of providers would enable them to expand the scope of their practice and the community-level health workforce more generally. 

Despite the reported lack of formal AMR training, most medical officers, nurses and animal health specialists could articulate their understanding of antibiotics and AMR. This is consistent with other studies in sub-Saharan Africa and similar contexts (see: [45], [46] and [47]) that reported providers’ familiarity with AMR and its main causes. For example, drug overuse or incorrect dosing was perceived as a leading cause of AMR. However, similar to other studies [48,49,50,51], in most cases there was uncertainty in the difference between viral and bacterial infections, alongside which illnesses antibiotics can treat that can lead to their use to treat viral infections, as reported elsewhere [50]. Community health providers and market/retail vendors evidenced the most variation on knowledge and understanding of AMR, viruses and bacteria. Yet, their importance in improving access to treatment and medical expertise has been widely acknowledged throughout the developing world [51,52,53]. They are therefore ideal candidates for tailored training to address potential gaps in understanding and improve the quality of their service and advice. As evidenced, however, in addition to knowledge gaps, they face significant challenges in being able to effectively provide care for their patients’/clients’ needs. Common constraints relate to generating income and procurement of supplies, including antimicrobials. This is compounded by the fact that they are private actors who operate outside of the public health system and are ultimately driven by profit, which may complicate their motivations to steward the use of antimicrobials.

Health providers often blamed the actions of patients and community members for the overuse and misuse of antibiotics. This is consistent with other studies with health providers in similar contexts [46]. Self-treatment and not finishing the prescribed dose were perceived as key drivers of AMR. Varied reasons were given for this. Respondents characterised patients and community members as simply being lazy and demanding (i.e., requesting specific medications). Others spoke negatively about their ability to follow instructions. In some cases (and also evidenced in other studies [54]), health providers blamed each others’ practices as the cause of AMR. Higher-tier health professionals cited over-prescribing or substituting medications in order to maximise profits (e.g., in the case of retail vendors). These narratives of blame [30] can be problematic, but also offer opportunities for improving doctor/patient communication. In addition, as discussed previously, these vendors experience their own challenges in providing health services. They are thus faced with difficult choices between balancing patient demand and prudent use of antibiotics. Interviewees also allocated blame to patients and clients for relying on traditional healers for health treatments, citing their lack of trust in biomedicine as reasons for this. To implement change, they advocated for biomedical healthcare providers and traditional healers to cooperate.

The focus on individual behaviours that contribute to AMR is rooted in a one-dimensional characterisation of the AMR threat that does not sufficiently account for the socio-structural forces underlying persistent suboptimal practices [55]. Many health providers recognised this and acknowledged that patients themselves experience numerous day-to-day constraints which impact their ability to stay healthy, access medications and use antibiotics responsibly. For example, health providers were aware that economic factors can lead people to self-treat without the advice of health experts, procure treatment from unofficial sources (e.g., neighbours), use leftover medicines, and fail to fulfil the prescribed dose. Actions to curtail the spread of AMR should be done in conjunction with Tanzanian regulators and strive to strengthen existing practices and policies aiming to prevent excess antimicrobial prescription and use without compromising critical access to antibiotics for those most in need.

Frustrations at the actions of individuals and their potential contribution to the spread of AMR obscure the connections between scales and systems that create the structures that perpetuate problematic practices around antibiotic prescribing and use [55]. Health providers often cited broader infrastructural problems, such as poor physical infrastructure and lack of adequate testing and drug procurement, as factors that inhibit their ability to steward antimicrobial use. These are well-recognised issues across sub-Saharan Africa and low- and-middle income contexts more generally [56,57]. However, solutions often focus on singular individual prescribing and consumption patterns. These wider structural challenges highlight the need to move beyond AMR mitigation strategies which are punitive at the individual level (“bad prescribers”, “good prescribers”), to an alternative approach that focuses on addressing the systemic challenges faced by institutions and the people working in them [55]. Until the barriers that prevent providers from acting upon knowledge are tackled, programmes that focus solely on provision of education are likely to fail [45,58,59,60]. Education alone or the prevention of prescribing certain antibiotics has indeed been evidenced as not being sufficient [61]. The prevention of AMR requires a more nuanced response that includes technology, research, political will, and leadership and responsibility [61]. More broadly, we need a multi-tiered approach to AMS that reorients focus away from individualised responsibility to collective action. This reorientation involves broadening lines of accountability beyond individual prescribers and consumers to include those often viewed as peripheral to the problem, for example, the executive and political classes, health funders, investors, and insurers, all of whom play an important role in perpetuating the structural conditions in which AMR is currently flourishing [55].

## 5. Study Limitations

Our data were collected from three communities with similar access to health facilities and programs; however, communities with greater/lesser access were not represented. This data are also solely of a qualitative nature, which may be limiting to some readers and in representing a broader scope of practice and experience around AMR. Furthermore, this study opted to analyse the experiences of animal and human health providers collectively, drawing out common constraints and challenges. We chose this approach because, while they work within different sectors, they operate within very similar material and contextual constraints (e.g., underfunded health systems, lack of infrastructure and lack of medical supplies). By analyzing these issues collectively, we reinforce the need for a One Health approach to consider the common constraints and challenges these providers face, instead of within their separate silos. Secondly, and relatedly, animal and human health are not considered wholly separated by the communities themselves. People’s livelihoods, health, and wellbeing are tied to livestock and livestock health and wellbeing in particular, and patterns of seeking treatment are often similar. This is also important in management and prevention of zoonotic diseases and other issues that may lead to antibiotic use. Furthermore, it is of value because of the crossovers in AMR through the use of livestock products, as well as the shared spaces of life. However, we recognize that this collective approach may miss the nuance of analysing animal and human data separately. This is an area that would benefit from further investigation in order to identify factors that can facilitate truly integrated working practices and policy between animal and human health sectors.

## 6. Conclusions

The need for strengthening knowledge on AMR and AMS among the health workforce has been highlighted globally. Here, we examined the opportunities and challenges facing health professionals in northern Tanzania in relation to AMR and AMS, focusing on settings representative of many highly-constrained and precarious environments. Health workers have a great motivation for contributing to healthcare and supporting their community, while needing to secure their livelihoods. However, they are not fully equipped with the formal training and qualifications they need to fulfil their roles. An especially neglected group of providers, both in terms of formal training and resources, includes community health providers and medical vendors, despite them providing critical access to healthcare in communities. Many health providers cite the actions of others as contributing to AMR, although they recognise that these are often driven by a lack of choice. They also report systemic constraints affecting their daily practices including staff shortages, inoperative drug procurement systems and unavailability of diagnostics. Overall, there is an eagerness for accessing training and for improvements in stewardship at the local level. The involvement of the broader AMR community in discussions of the issues identified will be essential to addressing the blame narrative pervasive in the AMR discourse, increasing awareness of the challenges that prevent optimal AMS, and reducing the structural factors constraining providers’ everyday practices.

## Figures and Tables

**Figure 1 antibiotics-12-00243-f001:**
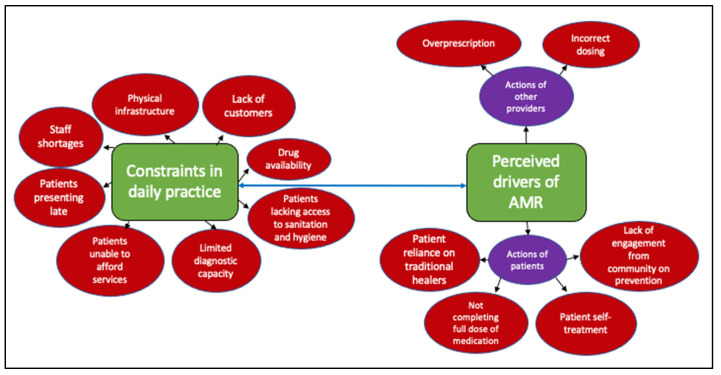
Drivers of AMR based on providers’ perceptions. These include those linked to the actions of others, patients/farmers or providers themselves, and those related to the constrained environments they operate within.

**Table 3 antibiotics-12-00243-t003:** Number of in-depth interviews (IDIs) conducted in three regions of northern Tanzania to discuss human and/or animal health.

Health Provider Type	Number of IDIs—Human	Number of IDIs—Animal	Location
Providers within health institutions	1	1	Kilimanjaro
	6	1	Mwanza
	2	1	Arusha
Retail/Market-based Providers	2	1	Kilimanjaro
	4	1	Mwanza
	1	3	Arusha
Community-Based Health Providers	3	5	Kilimanjaro
	4	6	Mwanza
	1	1	Arusha

**Table 4 antibiotics-12-00243-t004:** Human Health Actors’ Motivations for Working in Healthcare.

Level	Human Health Role	Description
Health institutions: Degree holders	Medical Doctor/Officer	Supporting their community to overcome- health problems. Interested in medicine.
Health institutions: Mixed post-secondary qualifications/training/knowledge	Clinical Staff and Nurses	Supporting their community
Retail/Market-based providers	Type II Drug Providers—*maduka la dawa muhimu/baridi* *; ADDO regulated	Livelihood support: many instances of nurses or clinical staff selling medicines to support their own income. Supporting their communities: passionate in helping communities. Interested in medicine.
	Other drug providers –general store, *maduka la kawaida*, not ADDO regulated	Interested in medicine
Community health providers	Traditional Healer	Supporting their communities: knowledge passed through families and in response to the demand for support
	Community Health Worker	Supporting their communities in having healthy lives and accessing their rights

* *duka la dawa* (literal translation “drug shop” refer to drug dispensing outlets) are also called “*duka la dawa baridi*” (literal translation “cold drug shops” referring to the drugs sold for non-severe conditions) and are now referred to as “*duka la dawa muhimu*” translated as “essential drug outlet”.

**Table 5 antibiotics-12-00243-t005:** Animal Health Actors’ Motivations for Working in Animal Health.

Level	Animal Health Role	Description
Health institutions: Degree holders	Veterinarian	Supporting their community: supporting animal healthcare development, training the next generation of animal health professionals interested in medicine
Health institutions: Mixed post-secondary qualifications/training/knowledge	Paraveterinarians (includes LFOs)	Supporting their community: caring for their animals Livelihood support: keeping animals healthy enables immediate access to income
Retail/Market-based providers	Agrovets/livestock medical vendors/veterinary centers/open-market livestock drug vendors	Livelihood support: responding to demand for animal medicines
Community health providers	Traditional Healer	Supporting their community: considered to be their calling. Desire to continue practice knowledge passed through family heritage.
	Community Animal Health Worker	Supporting their communities: knowledge passed through families and in response to the demand for support
	Expert Farmer	Livelihood support: liquidity in keeping and selling cattle; manure used to fertilize crops; milk Importance of demonstrating assets

## Data Availability

The datasets generated and/or analysed during the current study are not publicly available as the qualitative analysis is still ongoing. Identifying information may still be ascertainable from the raw interviews, but data summaries are available from the corresponding author on reasonable request.

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
