# Peer review of "“If You Do Not Take the Medicine and Complete the Dose…It Could Cause You More Trouble”: Bringing Awareness, Local Knowledge and Experience into Antimicrobial Stewardship in Tanzania"

_antibiotics, 2023, doi:10.3390/antibiotics12020243_

Round 1
Reviewer 1 Report
Overall the paper provides a useful and in depth examination of the challenges facing introduction of AMS (antimicrobial stewardship) programs in rural Tanzania. It appropriately reflects on the need to combine formal training with addressing the social, economic and health system factors that impinge on the ability of providers to apply training.
While this more holistic and system focused approach is welcome, the reader is left somewhat puzzled in regard to the purpose of the paper until the discussion section. An explanation of the purpose and thus to some extent the ‘operating context’ of the paper in the introduction and in the abstract would clarify this for the reader.
Abstract : No clear statement of the purpose / objective; and little description of the context (eg relevance of AMR in northern Tanzania). The final sentence obliquely references the role of training, but is not explicit on the purpose of the research, as mentioned at the start of the discussion section.
Introduction:
The introduction commences with fairly detailed description of the different types of providers in the Tanzanian health system. This is then followed by some more generic information on AMS and the Tanzanian national program.
I would suggest that the order here be reversed, with the more general contextual information preceding the more detailed presentation of the types of providers in the system.
Linkage to Tanzanian National Action Plan on AMR strengthens argument
Line 109 – incomplete sentence (starting with because)
Line 134 ff – provides some detail on the objectives of the study
Line 142 ff – useful acknowledgement of the need to understand the context of operation of the respondents
However, no clarification on the specific region / location of the study, and the characteristics / facilities in the region
What was the basis / rationale for selection of the 4 key areas/ factors: (line 134ff) eg from the literature; or from the experience of the researchers ?
Methods
This provides some information on the location of the study, but the ‘matching’ of villages (Line 154) is not clear ie does this mean the villages had similar levels of these factors, and if so, what were the levels of the factors eg population size, extent of access to health care etc.?
Results
Useful presentation of training, attitudes and context of interviewees.
Presentation of attitudes and practices in relation to use of antibiotics clear and well supported by quotations
Figure 1 useful summary of constraints and indicates broad scope of the constraints.
Useful detail on the specific issues in the quotations
Discussion
This commences with a statement of the purpose of the research, in regards to training. However, it would be much clearer to put this purpose statement in the Introduction, rather than leave it to the discussion.
The discussion summarises the results and the implications for training, and provides some comparison with the literature. The recommendations particularly in regard to community healthcare providers are especially relevant.
The comments (line 637 ff) in regard to the socio-structural factors that might influence behaviours are very pertinent.
Reviewer 2 Report
The manuscript reads as if it stems from a larger research initiative, of high quality and clearly of global importance, and recognising the interrelationships of AMR across different sectors.
I did not find bringing the interviews/results from the human and animal carers helpful. I agree entirely with the need for integrated working/practices/policy, but, unless the study looks at the facilitators to this, I found more confusion than a clarification of results and interpretation. This is especially given the different economic drivers/employment/career paths which might influence some points expressed.
The background section did not justify why the study was in northern Tanzania and only referred to east Africa. These places/territories are all very different.
I found the section on AMS (L99-126), the evidence critique to be very weak. It did not credit the excellent evidence of factors responsible for effective AMS, in different settings, including a number of high-quality reviews. Nor does it give reference to anything taking place amongst agriculturalists/veterinarians. A single sentence is only given to this, and there is more literature that could be drawn upon, and excellent evidence of how agriculture and veterinarians have responded well to changes in practice for AMR/AMS. Recognising the factors for this, in developed and less developed parts of the world is essential.
L128 - why is access to training constrained - do you mean just for CPD, or is there also an inference that qualified staff to have a lesser standard of undergraduate qualifications too?
Need to describe and give more justification for the study setting. What was special or similar about it to elsewhere and previous studies?
L150 - any reason why the data collection was broken into these two stages/periods?
Why 43 interviews? justification, and why 18 FGD, and why three villages
Why try and match the villages? What was the aim? Yet looking at Table 3, the information suggests matching didn't actually work/take place effectively.
L182 - what methods were used for translation? This is an extremely important factor to consider as part of the methodology for this study, including the analysis and interpretation. Therefore this section needs to include the methods used, any checks included, quality assurance and so on and then included as a potential weakness/issue in the Discussion.
Discussion -
L449 - I'm not sure if this is the primary purpose. Isn't it to identify contextual factors that need to be integrated within the design and delivery of training programmes? I think it would help to have clear aims and objectives in the Abstract and in the Methods to be certain throughout.
L554 - but it wasn't just representative of agricultural settings? didn't they consider human healthcare too?
L579 - arguably one might expect professionals to always say they want to help their community/moral obligations and so forth, and I've never come across a study that reports the opposite when these types of questions are put to health workers. Yet evidence exists that even with the knowledge and skills, prescribing antibiotics can include making money, or reduce time, or to keep customers happy, keeping repeat business, keeping a boss happy, and so on. Similarly, people often say they are interested in receiving more training, but do they take up any offers or why don't they? So in a way, the study is not doing anything to understand why this doesn't change, from the individual factors - what are the constraints in relation to these particular activities then, and how does it link with the constraints that were recognised/presented.
L604-607 - confusing to read, and seems to contradict itself. If they understood, then why find it hard to confusing to differentiate between bacterial and viral?
L649 poor sentence construction
The discussion also lacks a critique of the methods.
It needs to provide justification for having analysed the data as one, from the different health sectors, rather than human in one analysis, animal in another, and what potential problems this creates.
